

# Glacier dynamics across precipitation gradients: Can ice-elevation feedbacks overcome the rain shadow in the Olympic Mountains, USA?

Andrew A. Margason[1], Alison M. Anders[2], Robert J.C. Conrick[3], and Gerard H. Roe[4]

[1]Department of Geology, University of Illinois, Urbana, IL, 61801, USA; now at Ramboll 60606, USA
[2] Department of Geology, University of Illinois, Urbana, IL, 61801, USA
[3]Department of Atmospheric Sciences, University of Washington, Seattle, WA, 98195, USA
[4]Department of Earth and Space Sciences, University of Washington, Seattle, WA, 98195, USA

*Correspondence to*: Alison M. Anders (amanders@illinois.edu)

**Abstract.** Glacier extent is known to be sensitive to climate variability through time. The impact of spatial variability

in climate on glaciers has been much less studied. The Olympic Mountains of Washington State, USA, experience a

pronounced precipitation gradient with modern annual precipitation ranging between ~6.5 meters on the high west-facing

slopes to ~0.5 meters in the northeast lowlands. In the Quinault valley, on the west side of the range, a glacier extended onto

the coastal plain reaching a maximum position during the early Wisconsin Episode glaciation. There is no evidence of a large

Elwha glacier extending into the northeast lowlands at that time. We hypothesize that the asymmetry in past glacier extent was

driven by spatial variability in precipitation. To evaluate this hypothesis, we first constrain the past precipitation gradient, and

then model glacier extent. We explore variability in observed and modelled precipitation gradients over timescales from 6

hours to ~100 years. Across three data sets, basin-averaged precipitation in the Elwha is 54% of that in the Quinault, with

variability of less than 15% at the annual timescale. Specifically, this ratio does not consistently vary with regional climate

patterns. On average, modelled 6-hour accumulated precipitation in the Elwha is 78% of that in the Quinault during a winter

season, with a few low-precipitation time periods exhibiting a flatter or even reversed precipitation gradient. Overall, our

analysis does not suggest a mechanism for increasing the precipitation gradient, but overwhelmingly indicates spatially

coherent variability in precipitation across the peninsula. We conclude that the past precipitation gradient was likely similar to

the modern gradient. We use a one-dimensional glacier flowline model, driven by sea-level summer temperature and annual

precipitation to approximate glacier extent in the Quinault and Elwha basins. We find several equilibrium states for the



Quinault glacier at the mapped maximum position within paleoclimate constraints for cooling and drying, relative to today.

We assume the Elwha remained drier than the Quinault, and model Elwha extent for the climates of the Quinault equilibria.

At the warm end of the paleoclimate constraint (10.5˚C), the Elwha remains a small valley glacier in the high headwaters. Yet,

for the cooler end of the allowable paleoclimate (7˚C), the Elwha glacier advances to a narrow notch in the valley. As the ice

is forced to flow through a smaller cross-section, it thickens, triggering an ice-elevation feedback. This feedback leads to rapid

extension of the Elwha glacier to elevations only ~100 meters above those reached by the Quinault. While there is uncertainty

in the glacial record of the Elwha, it is unlikely that such a large glacier existed during the most recent glaciation. Therefore,

we suggest that the last glacial maximum climate was more likely to have been within the warm end of the paleoclimate range.

Alternatively, spatially variable drivers of ablation including differences in cloudiness could have contributed to asymmetry

in glacier extent. Future research to constrain past precipitation gradients and evaluate their impact on glacier dynamics is

needed to better interpret the climatic significance of past glaciation and to predict future response of glaciers to climate change.

## 1 Introduction

Glaciers are sensitive indicators of climate change through time because glacier extent is dependent on climate

variables including precipitation, temperature, insolation, and cloudiness (e.g., Braithwaite et al., 2003; Wagnon et al., 2003;

Kessler et al., 2006; Rupper and Roe, 2008; Anders et al., 2010). The sensitivity of glaciers to climate has been used to interpret

past changes and predict future changes in glacier extent as a response to, and record of, temporal changes in climate

(Oerlemans et al., 1998; Roe, 2011). However, climate not only varies in time, but also in space. Spatial variability in

precipitation due to topographic forcing (i.e., a rain shadow) is a well-documented and generally well-understood example of

a robust spatial gradient in climate (Roe, 2005). How does glacier extent vary across a mountain range with a significant rain



shadow? We consider this question for the Olympic

Mountains of Washington State, USA (Figure 1a). This mountain range on the Pacific northwest coast has small modern glaciers and evidence of extensive glaciation during the Pleistocene (Thackray, 2001). We begin by assessing the possibility of significant changes in rain-shadow strength

during glacial conditions relative to the present. Having established that spatial gradients in precipitation are likely a robust feature of the climate of the Olympic Mountains, we simulate the evolution of glaciers on the windward and leeward side of the range to examine the impact of the rain

shadow on glacier extent.

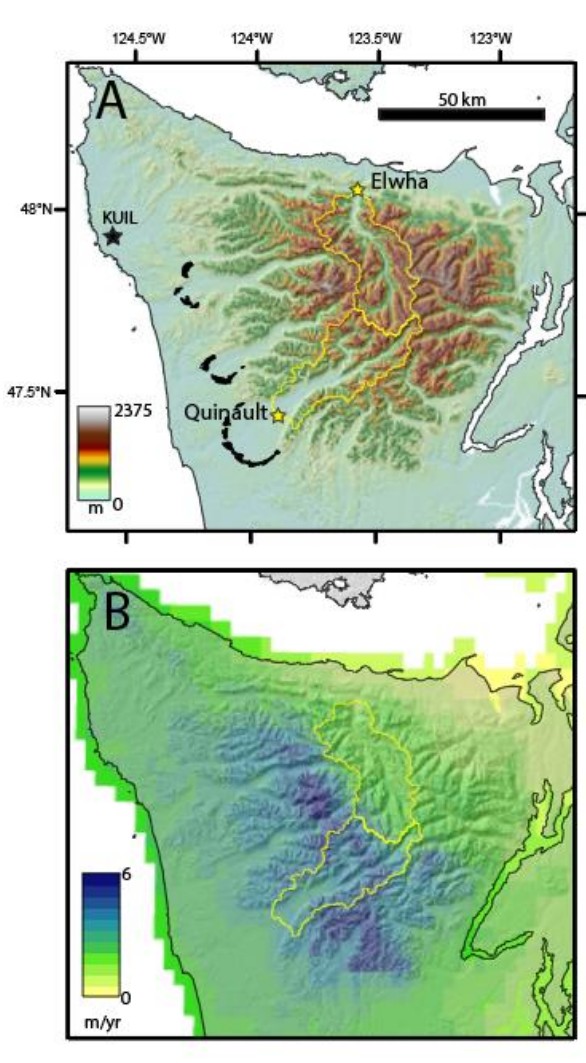

*Figure 1. Panel A: Topography of the Olympic Peninsula. Terminal moraines of the west side valleys shown in black (Marshall, 2013; Staley, 2015). Radiosondes are routinely*

*launched from the black star KUIL (Rutledge and Ciesielski, 2016). We define the Elwha and Quinault basins as the regions upstream of USGS stream gauges, shown as yellow stars. Panel B: Annual climatological precipitation from the PRISM model (Daly et al., 2008). Panel C: Total modelled*

*precipitation during the OLYMPEX campaign (Skamarock et al., 2008).*

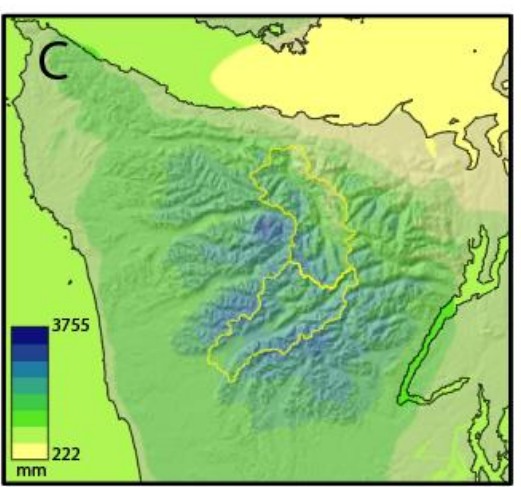

Modern precipitation across the Olympic Mountains is characterized by a steep precipitation gradient (e.g., Hobbs, 1978; Anders et al., 2007; Cao et al., 2018). Annual precipitation varies from ~6.5 meters at high elevations on the western side of the range to ~0.5 meters on the north-east lowlands (Figure 1b). On the Pacific coast, monthly precipitation totals during the drier warm season (May-September) vary from ~50-150 millimetres per month. In contrast, average monthly totals exceed 400 millimetres during November, December, and January (Minder et al., 2008). Large cool-season precipitation totals are due to the range lying in the path of the wintertime midlatitude storm tracks coming from the Pacific Ocean (Houze et al., 2017; Purnell and Kirshbaum, 2018). Winds directed into the topography induce standing waves in the atmosphere that cause lifting upwind of the topography. As air rises, it cools causing condensation of cloud droplets; eventually, their coalescence forms falling hydrometeors. These precipitation particles tend to reach the land surface on the windward side of the range, causing large precipitation totals at high elevations along the south-west side of the range. After passing over the crest of the Olympic Mountains, air descends and warms, causing evaporation of hydrometeors and cloud-water droplets. A significant fraction of the moisture has been removed from the air mass, leaving little potential for precipitation on the eastern side of the range (Zagrodnik et al., 2019).

The paleoclimate of the Olympic Mountains can be inferred through pollen data from a mire at Humptulips in the southwest of the Olympic Peninsula. The pollen data constrains past vegetation and climate approximately 30,000 [14]C yr B.P. (Heusser et al., 1999). The ecosystem during the time of the pollen deposition is interpreted as a coastal tundra dominated by pines. Sea-surface temperatures were more than 5°C colder than present, with precipitation close to 1000 millimetres per year, as compared with approximately 3000 millimetres per year today. We use these paleoclimate estimates of temperature and drying (relative to the modern) as a constraint on the climates we simulate in our glacier models.

Large glaciers repeatedly descended the western valleys during the early and middle Wisconsin glaciation ~40 kya. This formed U-shaped valleys with steep walls and terminal moraines beyond the range front on the coastal plains (Thackray 2001; Dragovich et al., 2007; Marshall, 2013; Stacey, 2015). In contrast, there is no clear evidence of alpine glaciation at low elevations in the Elwha valley on the leeside of the Olympic Mountains. Surficial geologic mapping of the Elwha and Angeles Point 7.5-minute Quadrangles approximately 60 km below the headwaters of the basin, did not recognize alpine till from an Elwha valley glacier (Polenz et al., 2004). Therefore, alpine ice in the Elwha valley was most likely quite restricted relative to

the west-side valleys (Hoh, Queets, Quinault). During the most recent glaciation, continental ice of the Puget Lobe of the Cordilleran Ice Sheet advanced into the Strait of Juan de Fuca 17-14 kya B.P. damming the mouth of the Elwha basin

(Easterbrook, 1986; Duda et al., 2011). This event could have obscured the geomorphic and surficial geologic record of earlier extensive alpine glaciation of the Elwha valley, allowing for some possibility of unrecognized extensive alpine glaciation in the basin.

There are no modern glaciers in the Quinault valley. However, there are modern alpine glaciers in the Elwha basin, limited to elevations between 1200 and 2200 meters (Riedel et al., 2015). This difference likely reflects the dissimilarities in

the hypsometry of the basins with the Elwha having greater area at higher elevation than the Quinault. More generally, it demonstrates that the sensitivity of glaciers to climate is not uniform across the Olympic Mountains. Instead, west-side glaciers and east-side glaciers may be sensitive to different ranges of climate variables.

The tectonic setting of the Olympic Mountains is an accretionary prism at a convergent plate boundary. The oceanic Juan de Fuca plate is subducting below the continental North American Plate. Sediments once deposited on the ocean floor or

in the subduction trench of the Juan de Fuca plate accumulated and were deformed during subduction into a thick wedge known as an accretionary prism (Tabor and Cady, 1978). The Olympic Mountains are the emergent portion of this accretionary prism. Subduction and deformation of the accretionary prism is ongoing. Thermochronology suggests that the rate and pattern of rock uplift in the Olympic Mountains has been consistent since ca. 14 Ma and has an average uplift rate of ~0.28 km Myr$^{-1}$ (Brandon et al., 1998). Exhumation of the Olympic Mountains varies in space from approximately zero at the coast to ~0.9 km

Myr$^{-1}$ in the core of the range, in a bullseye pattern (Michel et al., 2018).

The Olympic Mountains are hypothesized to be in both a flux steady-state and a topographic steady-state because rates and spatial patterns of erosion appear to match long-term rock-uplift rates (Pazzaglia and Brandon, 2001). Specifically, a flux steady-state requires that the material flux into the range (uplift) is balanced by a material flux out of the range (exhumation) so that the volume of the range is fixed in time (e.g., Willett, 1999; Adams and Ehlers, 2018). A topographic

steady-state is more restrictive and requires that the average cross-sectional shape of the range, and therefore average topographic cross-section is also fixed in time. The evidence for a topographic steady-state comes from the close correspondence of fluvial incision rates inferred from river terraces on the west side of the Olympics in the Clearwater Valley.

Fluvial-terrace incision rates agree with long-term exhumation rates derived from fission-track apatite ages in the same region

(Pazzaglia and Brandon, 2001). The matching rates and spatial patterns of erosion at the timescale of tens-of-thousands of

years with exhumation at the timescale of a few-million years supports the hypothesis that the topography has remained

consistent as material moved through the accretionary prism. However, fluvial processes have not been the rule in the past

million years. The Clearwater River is in the only substantial valley on the west-side that was not glaciated during the

Wisconsin Episode, and therefore did not experience conditions typical of the range for the last several tens-of-thousands of

years. A greater understanding of glacier extent and erosion could provide additional data to test the hypothesis of topographic

steady-state.

There is strong evidence in the Olympic Mountains for 1) the modern precipitation gradient, and 2) large glaciers that

were present on the windward side of the range but not on the lee side during the last glacial episode. Therefore, we hypothesize

that spatial variability in precipitation controlled past glacier extent. This points us to two questions. First, was the spatial

variability of precipitation in the past analogous to the modern variability? Second, does the precipitation gradient explain the

asymmetry in glacial extent on either side of the mountain range? To answer these questions, we analyse the modern climate

across short- and long-term time scales to see how variable the modern precipitation gradient is. We use this analysis of modern

variability to assess the likelihood of substantially different past precipitation gradients. After establishing the likely strength

of the past precipitation gradient, we use a 1-D flowline model to predict the extent of glaciation in the Quinault and the Elwha.

## 2 Spatial Variability in Precipitation

### 2.1 Methods

We characterize the spatial variability in precipitation across the Olympic Peninsula by comparing the amounts of

precipitation in the Quinault and Elwha basins. We define a rain shadow index ($R$) in Eq (1):

$$R = \frac{Pe}{Pq} \tag{1}$$


where *Pe* is the spatially averaged precipitation rate in the Elwha basin and *Pq* is the spatially averaged precipitation rate in the Quinault basin. We compute *R* over different timescales using three datasets: climatologies based on spatial interpolation of 30-years of observations, annual-precipitation totals inferred from river-discharge measurements, and 6-hour modelled precipitation outputs from a numerical weather-forecast model (the Weather Research and Forecasting, WRF, model).

By examining the variability of the rain shadow in time, we can compare it to factors that might influence the precipitation processes in this region. Specifically, we compare 6-hour measures of *R* with wind speed, wind direction, temperature, and atmospheric stability to assess the extent to which different types of precipitation events (e.g., cold vs. warm fronts) exhibit different characteristic rain-shadow strengths. At the annual timescale, we compare the annual timeseries of *R* derived from river discharge with climate indexes to see if a strong or weak rain-shadow correlates with different regional

climate states. Overall, our goal is to constrain the variability in the rain shadow over long and short timescales in the modern climate to understand causes of particularly strong and weak rain shadows. We leverage this data to constrain the rain-shadow strength during periods of intense glaciation of the Olympic Mountains.

### 2.1.1 Rain-shadow strength derived from spatially interpolated climatologies

The Parameter-elevation Regression on Independent Slopes Model (PRISM) uses observations from atmospheric monitoring sites to interpolate climate variables across the United States (Daly et al., 2008). The model divides terrain into topographic facets defined by slope and elevation. We use PRISM at a 4-km resolution normalized over a 30-year period from 1981 to 2010 (Figure 1b) and define a rain-shadow index, $R_{PRISM}$, as the ratio of the $P_e$ and $P_q$ calculated from PRISM model.

### 2.1.2 Rain-shadow strength derived from weather-forecast models

We use archived Weather Research Forecasting (WRF; Skamarock et al., 2008) from the Olympic Mountains Experiment (OLYMPEX; Houze et al., 2017) to understand the strength of the rain shadow at short timescales (Figure 1c). OLYMPEX was a meteorological field campaign in the winter of 2015 - 2016 designed validate Global Precipitation Measurements (GPM) satellites, and investigate Pacific midlatitude storms as they travel over the Olympic Mountains (Houze

Earth **Surface**
**Dynamics**
Discussions

et al., 2017). The WRF model was initialized using meteorological fields from the Global Forecast System and was run during

the OLYMPEX campaign (November 1st, 2015 to February 1, 2016). WRF parametrization and model setup are as described

in Conrick and Mass (2019a, 2019b).

We evaluate the variable RAINNC which is the cumulative total accumulation of ice and liquid precipitation at the

ground surface since the start of the model run. Specifically, 24-hour long simulations at 1.33km resolution were initialized at

00 and 12 UTC daily, with model output saved every 6 hours.  We create a time series of precipitation based on accumulation

during the first 12 hours of each model run. The first saved value of RAINNC provides a record of the 6-hr accumulated

precipitation during the first 6 hours. To get accumulated precipitation during the second 6 hours of each model run we subtract

the value of RAINNC recorded after 12 model hours from the value recorded after 6 model hours.  We treat this time series as

precipitation data without attempting to quantify model errors. We define a rain-shadow index, $R_{WRF}$, as the ratio of $P_e$ and $P_q$

obtained from the WRF output. We calculate an $R_{WRF}$ for the whole accumulated precipitation during OLYMPEX. In order

to characterize its variability in time, we also calculated $R_{WRF}$ for every six-hour period for which $P_e$ and $P_q$ both exceeded 5

mm.

The time series of 6-hour $R_{WRF}$ is compared to wind speed, wind direction, and temperature measurements collected

by weather-balloon radiosondes during the OLYMPEX campaign with the goal of identifying relationships between rain-

shadow strength and atmospheric conditions. Radiosonde data were taken at KUIL (Figure 1b). KUIL, located in Quillayute,

WA had 162 soundings from October 29, 2015 to January 16, 2016. Soundings were collected at variable heights, and we

interpolated these to a defined spacing of 250 meters using linear interpolation. We averaged temperature, wind speed, and

wind direction over an arbitrary one-kilometre vertical distance near the surface from 750 to 1750 meters for each sounding.

These average values were interpolated from the times of observation to 6-hour time intervals of the WRF model using linear

interpolation.

### 2.1.3 Rain-shadow strength derived from river-discharge records

River-discharge data collected at gauges in the Quinault and Elwha rivers (Figure 1b) were obtained from the USGS

(https://waterdata.usgs.gov). Discharge data was given as monthly averages measured in cubic feet per second. Complete data



from both basins was available from 1919 to 2019. We use river-discharge data to approximate the rain-shadow strength at

two timescales: the entire period of record and each water year (October to September). We define $E_{\mathrm{run}}$ as the total runoff

(i.e., discharge) divided by the area of the Elwha basin, and report the result in mm to compare with our precipitation estimates.

$Q_{\mathrm{run}}$ is the same metric for the Quinault basin. We then have a gauge-based estimate of the rain-shadow index:

$$R_{GAUGE} = \frac{E_{\mathrm{run}}}{Q_{\mathrm{run}}} \qquad (2)$$

For each of the 101 water years in the record we calculate water-year averages of $E_{run}$, $Q_{run}$, and $R_{GAUGE}$.

        We compared $E_{run}$, $Q_{run}$, and $R_{gauge}$ to climate indices. Monthly climate indices came from Climate Explorer

(https://climexp.knmi.nl) and characterize recurring patterns of atmospheric circulation. These patterns influence the formation

of mid-latitude storms and are also associated with significant temperature anomalies (e.g., Robertson and Metz, 1990;

Michelangeli et al., 1995). The indices used here are derived from measurements of interannual sea-surface temperatures in

the Pacific Ocean. We included four indices: the El-Nino Southern Oscillation (Nino 3.4; available from 1854-2021); the

Pacific Decadal Oscillation (PDO; available from 1900-2021); the Pacific North American Index (PNA; available from 1950-

2020); and the Tropical/Northern Hemisphere (TNH; available from 1950-2020). Each index was included because of its

influence on the Northern-Hemisphere extratropics and the Pacific Northwest in particular. (Barnston and Livezey, 1987).

These indices were averaged over December, January, and February (DJF) which is the period of greatest snowpack

accumulation (Serreze et al., 1999; Robertson and Ghil, 1999). We normalize the time series of $E_{run}$, $Q_{run}$, and $R_{GAUGE}$ by

their standard deviations and calculate Pearson's correlation coefficients (*r*) between these time series.




## 2.2 Rain-Shadow Results

### 2.2.1 Long-term measurements of rain-shadow strength

The PRISM climatology (Figure 1b) yields $R_{PRISM} = 0.54$ (Table 1), or approximately 85% more precipitation in the Quinault basin than the Elwha. We expect that there are significant local uncertainties in the PRISM climatology as it depends on the quality and spatial density of gauges used in the regression (Cao et al., 2018). 37 weather stations were used across the ~3700 km$^2$ Olympic Peninsula to derive the PRISM climatology, and only 3 sites of the 37 stations have an elevation above 500 meters. In the Coweeta basin in the southern Appalachian Mountains, local annual-mean errors in PRISM errors estimated at 16%, given 25 observation stations within the ~20 km$^2$ basin (Daly et al., 2017). In comparison, the lack of measurements of precipitation, especially at high elevations, in the Olympic Mountains suggests that errors significantly more than 16% are likely. Specifically, the lack of gauges in the Elwha headwaters means that PRISM does not measure spillover precipitation that crosses the divide to fall in the upper Elwha basin.

| Measure of $R$ | Source | $R$ (Standard Deviation) | Period of Record |
| --- | --- | --- | --- |
| $R_{PRISM}$ | PRISM | 0.54 | 30-year climatology (1981 - 2010) |
| $R_{GAUGE}$ | USGS River Discharge | 0.54 <br><br> 0.54 (0.14) | 1919 – 2019 <br><br> Individual water years |
| $R_{WRF}$ | OLYMPEX WRF-ARW | 0.72 <br><br> 0.80 (0.36) | OLYMPEX campaign <br> 11/01/2015 - 02/01/2016 <br> 6-hour periods with precipitation |

**Table 1.** Measures of R from PRISM, river discharge, and WRF datasets.

Earth **Surface**
**Dynamics**
Discussions

$R_{WRF}$ for the entire OLYMPEX campaign equals 0.72 (Table 1; Eq. (3), Figure 2). An $R$ closer to one means precipitation is more similar in the Quinault and Elwha. WRF forecasts predicted significantly larger amounts of spillover precipitation in the Elwha basin than the PRISM normal. However, the WRF forecasts are imperfect: WRF underestimated

precipitation by up to 100 mm at OLYMPEX gauge locations around the Quinault basin during larger storms (Conrick and Mass, 2019a). Additionally, the OLYMPEX campaign only studied a single wet season which may not have been representative of longer-term average precipitation.

A 101-year river discharge of the Elwha and Quinault basins yielded $R_{GAUGE}$ = 0.54 (Eq.5, Table 1), which is in agreement with $R_{PRISM}$. Uncertainty in discharge measurements and hence $R_{GAUGE}$ can stem from several sources. For

example, there is uncertainty in the measurements of the cross-sectional area of the flow and in the stage-discharge relationship. Additionally, river discharge represents water flowing off the land surface, neglecting the possibility of evapotranspiration of precipitation. The cool, wet climate of the Olympic Peninsula makes this error likely to be relatively small, but there could be systematic differences between the Quinault and Elwha in the conversion of precipitation to river discharge.

**2.2.1 Temporal variability in rain-shadow strength**

In addition to the average rain-shadow indices estimated using these three independent data sets, we also considered the variability in rain shadow index at timescales of hours and seasons. The six-hourly $R_{WRF}$ values have a mean of 0.80 and standard deviation of 0.36 (Table 1). The mean rain-shadow index measured in this way is larger than the OLYMPEX-total $R_{WRF}$ (=0.72), meaning that the 6-hourly forecast precipitation is more uniform across the mountain range than other data sets.

We note that at a 6-hour resolution it is possible that the measured rain-shadow index could reflect differences in timing of precipitation in the two basins. For example, if the heaviest precipitation in the Elwha occurs after it happens in the Quinault, then, the rain shadow index might be larger in one 6-hour period and smaller in the subsequent 6-hour period with neither measure reflecting the storm event. Therefore, we averaged WRF forecasts over longer time periods to produce time series of $R$ over a range of timescales. Over longer time periods, the mean value for $R$ and the standard deviation of $R$ gets smaller up

to 48-hours (Table 2). For the longest time period examined, 72-hours, the standard deviation increases relative to 48-hours,





possibly reflecting less coherence in conditions at these longest timescales. For all timescales examined, however, $R$ derived from WRF forecasts during OLYMPEX is larger than in other data sets. The larger $R$ may reflect a tendency of WRF to overpredict spillover precipitation in the Elwha basin. However, we do not have sufficient data to conclude that this is necessarily the case. Moreover, PRISM is not informed by data collected near the drainage divide and so is not necessarily

reliable. Similarly, $R$ estimated from differences in river discharge is complicated by the existence of difference in evapotranspiration between the basins, if this is the case. Finally, it is possible that the OLYMPEX period was anomalous compared with longer-term records.

| Time period of measurement | $R$ (Standard Deviation) |
|---|---|
| 6-hours | 0.80 (0.36) |
| 12-hours | 0.76 (0.30) |
| 24-hours | 0.74 (0.31) |
| 48-hours | 0.75 (0.26) |
| 72-hours | 0.76 (0.33) |

**Table 2.** Mean and standard deviation of $R_{WRF}$ at different time intervals for accumulated precipitation.

What causes the time variability of the 6-hourly $R_{WRF}$? Figure 2 compares the 6-hourly $R_{WRF}$ to the average amount of precipitation. It shows that the rain-shadow strength for large events is less variable than for all rain events. Specifically, 6-hour periods with average precipitation totals above 20 mm tend to cluster at $R$ values between 0.5 and 1.0, indicating that

higher values of 6-hourly $R_{WRF}$ are generally associated with lighter precipitation.

Earth **Surface**
**Dynamics**
Discussions

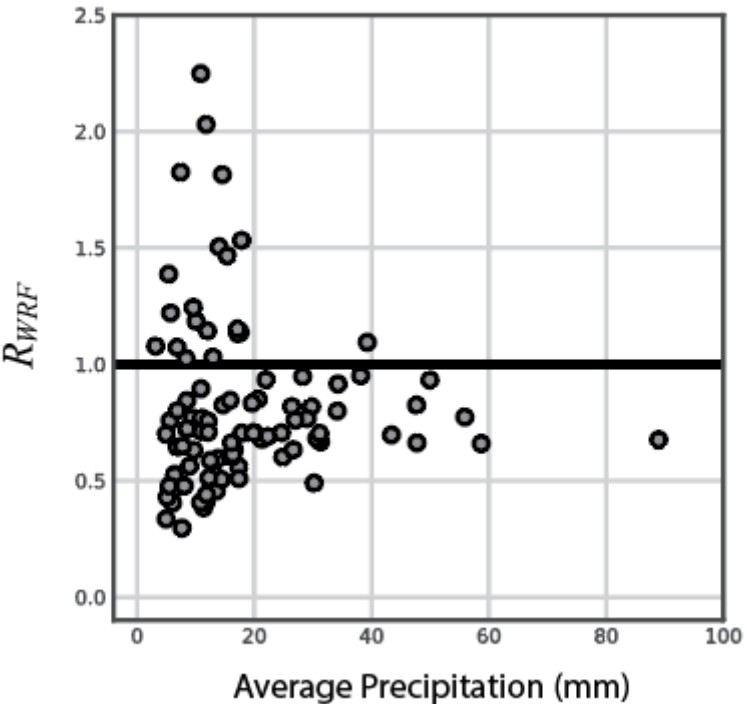

Figure 2. $R_{WRF}$ measured at 6-hour intervals varies with the total precipitation averaged over the two basins. As average

precipitation increases, $R_{WRF}$ becomes less variable.

We analysed the time series of 6-hourly $R_{WRF}$ to examine the relationships between R magnitude and temperature,

wind speed, and wind direction (as measured by atmospheric soundings) (Figure 3). The time series can be divided into groups

of low (0.34-0.69), moderate (0.70-0.99), and high (1.0-2.5) R. High R values (greater than one) were commonly observed

during colder weather conditions (Figure 3a). On the other hand, lower R values were associated with warmer temperatures.

High R events had lower magnitude precipitation compared to moderate R events, with rates of approximately 14 millimeters

per 6-hour interval and 26 millimeters per 6-hour interval, respectively. The wind speed during cold, high-R events was slower

compared to low- and moderate-R events (Figure 3b), while wind direction showed no significant pattern among the different

R groups (Figure 3c). In general, the temporal variability of $R_{WRF}$ suggests that colder conditions and slower winds lead to




Earth **Surface** Open Access
**Dynamics**
Discussions
EGU

lower magnitude and more uniform precipitation, while warmer and windier conditions are linked to R values close to the

long-term average. Small magnitude precipitation events with R values below 0.5 occurred during warm and windy conditions.

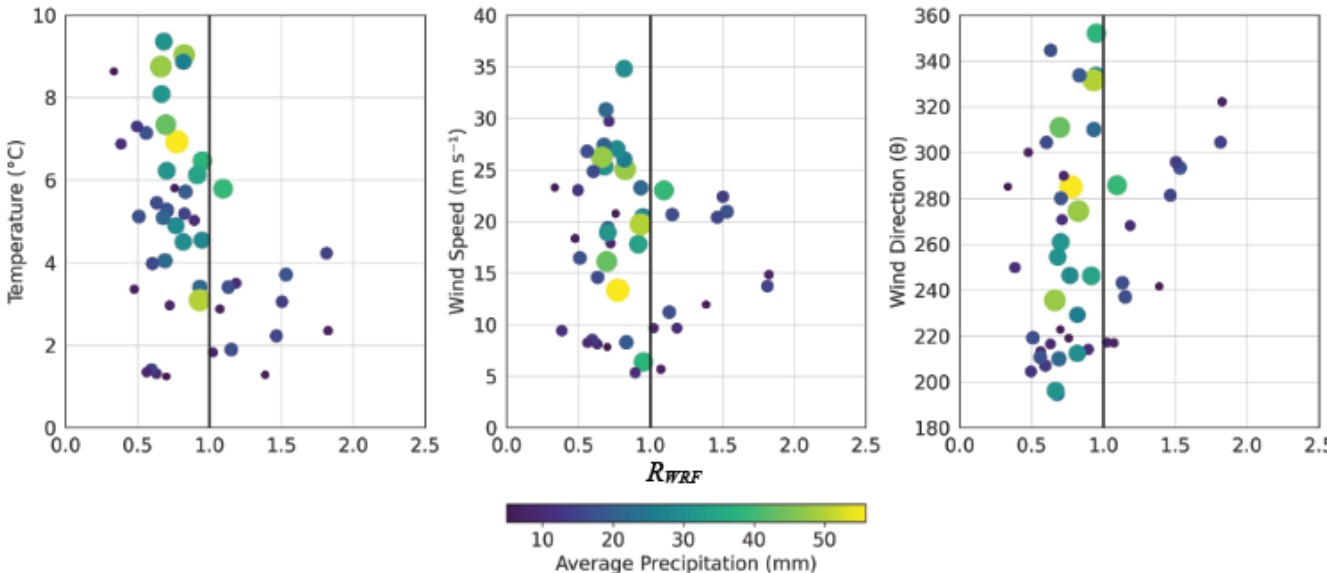

Figure 3. The relationship between $R_{WRF}$ at 6-hour resolution, and various atmospheric variables as measured by the KUIL
radiosonde. The size and colour of the data points correspond to the average precipitation across both basins.

We used discharge records to examine differences in rain-shadow strength in different water years. The annual time

series of $R_{GAUGE}$ has a mean value of 0.54, with a standard deviation of 0.14 (Table 1). We compared the time series of $R_{GAUGE}$,

$E_{run}$, and $Q_{run}$ to climate indices to determine if temporal variability in $R$ is correlated with specific climate patterns, for

example, La Niña vs. El Niño. Instead, we find that there is a remarkably consistent relationship between $Q_{run}$ and $E_{run}$ ($r =$

0.98) and the climate indices do not strongly correlate with $E_{run}$, $Q_{run}$ or $R_{GAUGE}$. (Figure 4). Our analysis demonstrates that

river discharge varies synchronously on the windward and leeward sides of the mountain range. This suggests that topography,

and possibly differences in evapotranspiration, strongly influence the rain shadow strength at the timescale of entire wet

seasons.






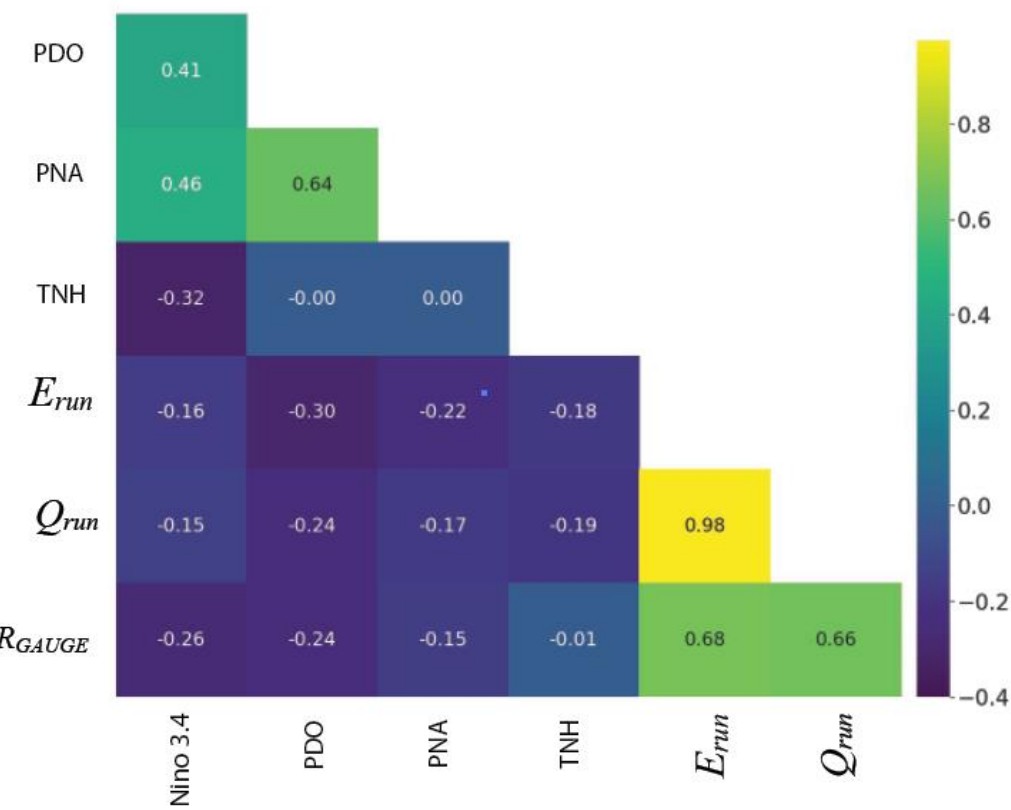

Figure 4. A correlation heat map displaying the relationship between various climate indices and $E_{run}$, $Q_{run}$ or $R_{GAUGE}$. $E_{run}$ and $Q_{run}$ are highly correlated with each other and $R_{GAUGE}$ is not correlated with climate indices.


### 2.3 Inferences for reconstruction of past climate

Three independent data sets give us $R$ of ~0.5 to ~0.7 with less variability in $R$ at longer timescales. At the shortest timescales examined (6-hours), the largest deviations in $R$ are values larger than 1 occurring during periods of cold, slow windspeeds, and light-precipitation. There is no example of a large precipitation event with $R$ much less than 0.5 during the

OLYMPEX campaign. At the timescale of water years, $R$ is uncorrelated with regional climate indices and river discharge

varies synchronously between the Quinault and Elwha basins. The lack of variability in $R$ at the water-year timescale, the lack of correlation of $R$ with climate indices, and the small number and magnitude of low $R$ events at the 6-hour time scale all suggest that a conservative minimum estimate of $R$ in the Olympic Mountains during the last glacial period is 0.5. The most likely difference between the modern and glacial climate suggested by our data would be an increase in $R$ (i.e., a more uniform

climate) due to cooler conditions (Figure 3).  We proceed, therefore, to explore how the current rain-shadow strength would have impacted glacier dynamics on the Olympic Peninsula.

### 3.  Modelled Glacier Extent in the Quinault and Elwha basins

### 3.1 Glacier Flowline Model Methods

We use a flowline model based on that of Oerlemans (1997) adapted from Roe and O'Neal (2009) to approximate the

dynamics of glaciers in the Quinault and Elwha basins. The model is one-dimensional but considers the effect of changes in valley width in considering mass conservation along the flowline. We follow the work of Roe and O'Neal (2009) by calculating the mass balance along the flowline as a function of precipitation and temperature, instead of imposing a specified vertical mass-balance gradient.  Inputs to the model include valley geometry, initial ice thickness, sea-level temperature, atmospheric lapse rate, and precipitation rate. The horizontal grid spacing in the model is 500 m.

Valley geometry is idealized as a long profile with a valley floor of defined width and valley sidewalls sloping outward at a fixed angle above the valley floor.  The Quinault and Elwha basins have different morphology and hypsometry (Figure 5) which is likely to affect glacier dynamics. For simplicity, the geometry of each basin was idealized using three segments with uniform width and valley angles for each segment with the valley-floor elevation defined by the elevation of the lowest part of the valley extracted from a 30m digital elevation model (Table 3).  We make no attempt to remove glacial and post-glacial

sediment from the valleys before extracting the long profile. Where the valleys are wide, the valley side walls are assumed to be vertical. Within narrow sections we define the angle at which valley walls slope outward. Both valleys start with a wide contributing area in the high headwaters. In the middle of each basin, there is a narrow notch. Below the notches, the basins open into expansive coastal plains. Basin widths were chosen to approximate upstream area accumulation along the long profile.

Earth **Surface**
**Dynamics**
Discussions


Figure 5: Panel A shows the Quinault and Elwha basins with topography represented as a grey hillshade and the 30-year climatological annual precipitation in mm yr$^{-1}$ shown by the colour bar. Red (Elwha) and Cyan (Quinault) lines trace the path of the flowlines modelled. Grey-shaded boxes are regions where the valleys narrow. Panel B shows the elevation and precipitation along these flowlines with the coloured lines showing the data from the digital elevation model (DEM) and

PRISM model and the black lines showing the approximate values used in our model.



| Basin Segment | Elevation Range (m) | Valley Width (km) | Valley sidewall angle (°) |
|---|---|---|---|
| Upper Quinault | 500-115 | 16.0 | 0 (vertical) |
| Middle Quinault | 115-60 | 2.5 | 60 |
| Lower Quinault | 60-0 | 18 | 0 |
| Upper Elwha | 1600-450 | 11.4 | 0 |
| Middle Elwha | 450-70 | 5.2 | 50 |
| Lower Elwha | 70-0 | 15.0 | 0 |

Table 3: Elwha and Quinault valley geometry parameterization for the one-dimensional flowline model.


Mass balance of the glacier varies along its length as a function of spatially variable precipitation and temperature. Temperature ($T$) varies along the length of the flowline ($x$), and is a function of elevation of the ice surface and an imposed sea-level temperature, $T_{SL}$:

$$T(x) = T_{SL} + \gamma\big(zb(x) + h(x)\big) \tag{3}$$


where $\gamma$ is the atmospheric lapse rate, here chosen as -8.5e-3 Km$^{-1}$, $zb(x)$ is the elevation of the bed and $h(x)$ is the thickness of the ice. Precipitation decreases down valley in both basins, and is generally lower in the Elwha basin (Figure 5). We idealize this variability by setting a precipitation rate at the top of the basin and imposing a fixed fractional decrease in precipitation from the top to the bottom of each valley. For the Elwha we assume precipitation at the valley mouth is 27% of that of the valley top. For the Quinault, we assume precipitation at the valley mouth is 44% of that at the valley top. These fractions were chosen to approximate the modern spatial pattern in precipitation (Figure 5). To simulate a glacial climate, we make the climate drier than modern by imposing a glacial drying factor (*GDF*) less than 1. The idealized modern precipitation gradient is multiplied by the *GDF* to model a drier climate that maintains the down-valley precipitation gradient observed today. We



compare modelled glaciation of the Elwha and Quinault basins with the same $T_{SL}$ and *GDF* to assess the response of the different basins to the same climate forcing.

       Modelled temperature and precipitation are used to calculate mass balance along the flowline. Precipitation is assumed to be adding mass throughout time and across the entire glacier surface. Temperature dependent melting is assumed to be the only loss of mass and is calculated as:


$$melt(x) = maximum\big(0, \mu T(x)\big) \tag{4}$$

where $\mu$ is a constant melt factor set to 0.65 m yr$^{-1}$C$^{-1}$. From this we calculate a spatially variable mass balance, *b(x)* in m yr$^{-1}$, as:


$$b(x) = P(x) - melt(x)$$

(5)

       With mass balance calculated, we follow the method of Oerlemans (1997) as implemented by Roe and O'Neal (2009))

and Stuart-Smith et al. (2021) to solve the nonlinear diffusion equation governing the height of the ice over time. We choose a fixed value of 1.9e-24 Pa$^{-3}$ s$^{-1}$ as the parameter controlling internal deformation and 5.7e-20 Pa$^{-3}$ m$^2$ s$^{-1}$ as the sliding parameter, based on values used in Oerlemans (1997). The model requires ice to be present at the beginning of simulations. Therefore, we start with glaciers of the expected scale and use a large glacier near the terminal moraine for the initial condition for the Quinault basin and use a small initial glacier limited to the headwaters in the Elwha basin. The model is implemented

in    Matlab    and    the    model    code    and    basin    geometry    input    files    are    publicly    available    at https://www.hydroshare.org/resource/a908fe06d4784684b68452d21efae69b/.



### 3.2 Results of glacier flowline modelling

We begin by finding the range of climate conditions that produce a modelled Quinault glacier in equilibrium at the

observed last glacial maximum extent (Figure 1a). Through trial and error, we find pairs of $T_{SL}$ and glacial drying factor (GDF) that produce an equilibrium glacier at the terminal moraine. $T_{SL}$ and *GDF* linearly trade off (*GDF = 0.10 $T_{SL}$ −0.55*) within the range of sea level temperatures between 7 and 11°C supported by paleoclimate reconstructions (Heusser et al., 1999). For a chosen value of $T_{SL}$, we can find the percentage of precipitation (*GDF*) that forces the glacier to stay in equilibrium near its terminal moraines over a 5,000-year period. Encouragingly, the corresponding *GDF* values (~0.15-0.55) are similar to the

required drying of at least 0.33 implied by the paleoclimate record.

Having found equilibria for the Quinault glacier, we use the same climate $T_{SL}$ and *GDF* pairs to simulate the evolution of the Elwha glacier. Pairs of climate variables that produced nearly identical equilibrium glaciers in the Quinault led to enormously different glaciers in the Elwha basin. At the hotter end of the climate range at 10.5°C the Elwha glacier reaches a steady state with a high-elevation terminus consistent with the most likely true behaviour of the Elwha glacier during the most-

recent glaciation. However, for a cooler climate at 9.5°C the Elwha glacier advances far down valley onto the coastal plain (Figure 6). The Elwha glacier advances from the headwaters into the narrower portion of the valley about 25 km from the divide. The glacier thickens due to the abrupt change in cross-sectional area. The thickening causes the surface of the ice to be at higher elevations. This increases the accumulation area of the glacier. A larger accumulation area causes glacier growth and further increases in glacier elevation. The occurrence of this ice-elevation feedback for the Elwha glacier is extremely sensitive

to climate conditions across a range of climate conditions that produces little change to the Quinault glacier.

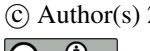


Earth **Surface**
**Dynamics**
Discussions



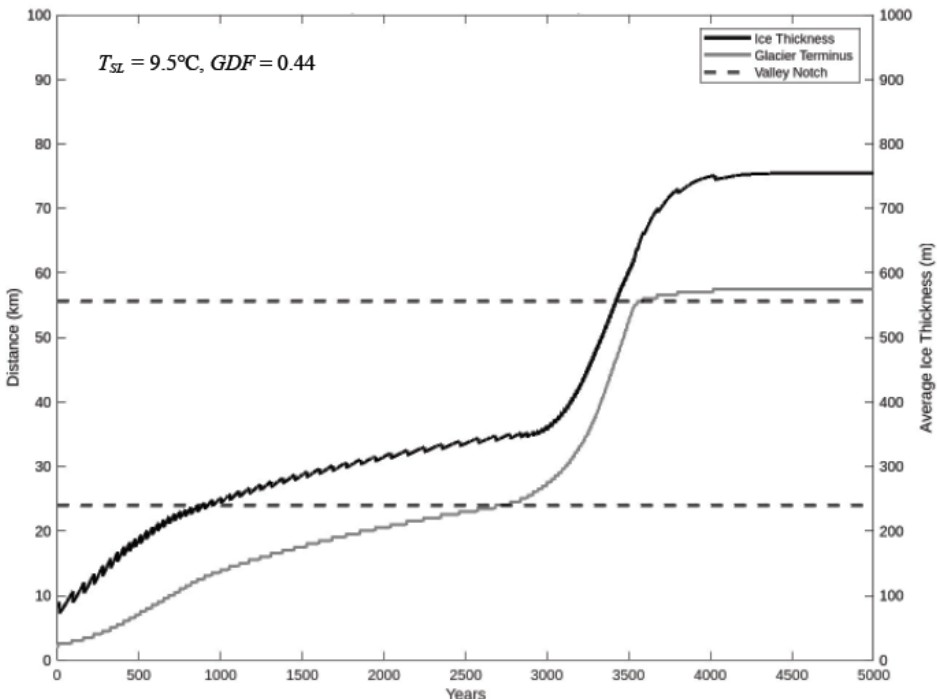

Figure 6. The temporal evolution of the Elwha glacier terminus and ice thickness for $T_{SL} = 9.5°C$ and $GDF = 0.44$. These

climate factors yield an equilibrium Quinault glacier at its terminal moraine position. As the Elwha glacier terminus (grey line)

advances into the narrow section of the valley (distance of approximately 23 to 56 km shown on left axis) just before 3000

years into the simulation, the ice thickness (black line) increases rapidly, initiating an ice-elevation feedback. Upon reaching

the other side of the valley notch, the glacier reaches a state of equilibrium, where the rates of accumulation and ablation are

roughly equal. It is important to note that this ice-elevation feedback is not observed in all climate simulations, but is more

likely to occur in cooler and drier conditions, as well as in basins with valley geometries similar to that of the Elwha.

Earth **Surface**
**Dynamics**
Discussions

### 3.3 Discussion of flowline model results

We observe that the Elwha glacier model experiences an ice-elevation feedback that is associated with the geometry of the basin. The initiation of the feedback occurs when the glacier advances into the narrow section of the valley (notch) where it begins to thicken (Figure 6). The Quinault basin also has a narrow segment of the valley in the middle of the basin; therefore, we expect to see a similar ice-elevation feedback occurring in this valley. To investigate this possibility, we ran a limited set of models with an initially small Quinault glacier. In some simulations, this glacier grew large enough to advance through the narrowing of the valley. It thickened and extended, demonstrating an ice-elevation feedback similar to that we observed in the Elwha simulations. However, a limited search of parameter space did not reveal any equilibrium state for a small Quinault glacier. When we begin the simulation with a small glacier in the headwaters of the Quinault valley it either melts completely or advances through the narrowing of the valley to become a large glacier. We conclude that the higher elevation headwaters of the Elwha allow a smaller equilibrium glacier which is not possible in the idealized Quinault basin geometry which extends only to ~600 meters elevation, while our reconstructed ELA is ~1000 meters.

The geometry of the basins is not the only control on the glacier extent. We ran a set of simulations switching the basin geometry. That is, we ran the Quinault climate over the Elwha basin geometry and vice versa. The Elwha glacier was consistently large when given Quinault precipitation rates. The Quinault glacier ablated away or grew large when given the Elwha precipitation rates. This indicates that the contrast in precipitation amount, rather that the basin morphology, is the main cause of different behaviour of the Quinault glacier and Elwha glacier in our models.

If we assume that geologic mapping of the Elwha basin has correctly identified the limited extent of recent glaciation, we can interpret our results as demonstrating that precipitation gradients are sufficient to explain glacier asymmetry if the past climate was at the warm and wet end of the range of paleoclimate estimates. However, the basin geometry and sensitivity of the Elwha glacier to climate variability within a plausible range suggests that large-scale glaciation of the basin during the Quaternary must be considered as a possibility.

In our model we have neglected other spatially variable surface processes that could have influenced the extent of the Elwha glacier. Specifically, we did not account for the separate influences that precipitation-type, cloudiness, humidity, and sublimation would have on glacier accumulation and melt. In the model, we did not discriminate between different types of

Earth **Surface**
Dynamics
Discussions

precipitation. All precipitation was considered snow. Rain has a significant impact on glacial melt rates and could influence

the mass balance of the glaciers (Oltmanns et al., 2019). The larger precipitation totals in the Quinault suggest that this effect

could be more important for this glacier than for the Elwha. Cloudiness modulates the amount of solar radiation that reaches a

glacier's surface. We expect that the same physics that creates the rain shadow should produce coherent special patterns in

cloudiness in the Olympic Mountains. Clouds condense on the windward side of the range due to orographic lifting, and then

evaporate when they descend on the leeward side of the mountain range (Roe, 2005). During some precipitation events, clouds

are observed to decrease in elevation and water content rapidly near the divide between the Quinault and Elwha basins

(Zagrodnik et al., 2019). If the Elwha valley is less cloudy than the Quinault, we expect that more solar radiation would increase

local air temperatures and enhance glacier melt (Huss et al., 2009). Additionally, there could be spatial patterns in humidity

that influence glacier mass balance. If the Elwha valley is systematically less humid, we expect that the lapse rate should be

larger than in the wetter Quinault.

We conducted a small number of experiments to determine the sensitivity of the modelled Elwha glacier to our choice

for the melt factor.  For the simulation shown in Figure 6, if we increase the melt factor slightly from 0.65 m yr$^{-1}$ K$^{-1}$ to 0.75

m yr$^{-1}$ K$^{-1}$ the Elwha glacier remains small.  For reference, the melt factor measured at various modern glaciers and ice sheets

shows variability of at least a factor of 2-3 (Braithwaite and Yang, 2000). The melt factor for Blue Glacier, on the west side of

the Olympic Mountains, was estimated using glacier area change and PRISM-based temperature and precipitation data

(Fountain et al., 2022).  Field measurements could constrain the melt factor in the eastern Olympics to assess the potential for

enhanced melt in the Elwha relative to the Quinault. Future work could establish if climatological variability in cloudiness is

equivalent to this degree of difference in the melt factor relative to the Quinault. A role for lapse-rate variability could also be

explored.

While we defend our analysis of the modern climate and our reasonable assumption that the past spatial gradient in

precipitation in the Olympics was similar to that observed today, it is possible that precipitation events during the last glacial

do not have analogs in the modern climate. One way to assess this possibility would be to use a forecasting model like WRF

with boundary conditions pulled from global circulation models of the last glacial maximum. These simulations could reveal

different mechanisms of precipitation variability than those that operate today.

Any assessment of erosion by glaciers on the Olympic Peninsula requires a clear picture of glacier extent through time. We could model glacial erosion in the Quinault basin by simulating the advance and retreat of the Quinault glacier through a glacial cycle and integrating basal sliding. The spatial pattern of glacier sliding within the Quinault could be

compared to gradients in exhumation across the range and fluvial erosion within the Clearwater valley.  Glacial erosion of the Elwha valley depends on the extent of ice. If the Elwha glacier remained small during the last glacial episode, then glacial erosion was limited and perhaps fluvial erosion was a stronger control on landscape-scale erosion on the lee side of the range. However, if an expansive and thick glacier ever occupied the Elwha valley, our models suggest it would have been very fast flowing and erosive – more so than the Quinault glacier.

Future field work could improve understanding of the past extent of the Elwha glacier. One target would be to find terminal moraines associated with the Elwha valley glacier during the most recent glaciation, presumably located up valley from the mapped quadrangles near the valley mouth. If these are above the notch, they would improve our confidence in the model simulations. In the lower Elwha valley, drill cores could be recovered and analysed to identify the extent of locally derived (alpine) till. It is possible that alpine till from previous glaciations could be identified from the mineralogy and

chemistry – allowing us to constrain past glacier extent.

## 4 Conclusions

We set out to understand the factors influencing the asymmetry in glacier extent between the Quinault and Elwha basins of the Olympic Mountains. To this end, we characterized the modern precipitation patterns, and assessed the potential

influence of various climate indices on these patterns. We calculated ratios of average precipitation in the Elwha basin relative to the Quinault basin using three different datasets: PRISM data, river discharge data, and WRF model output. We conclude that the long-term ratio of average precipitation between the two basins likely remained at-or-above 0.50 (<100% more precipitation in the Quinault than the Elwha) during recent glaciations.  We find no evidence of atmospheric or climatic influences, such as weather regimes, that would cause a more severe precipitation gradient in a glacial climate. Therefore, we

make the conservative assumption that modern climate is a good representation of past spatial gradients in precipitation.

Earth **Surface**
**Dynamics**
Discussions

To better understand the response of glaciers in the region to climate, we employed a one-dimensional glacier flowline model simulating the response of glaciers in both basins to a range of climate states. This included variations in sea level temperature ($T_{SL}$) and the glacial drying factor ($GDF$). There exists a set of $T_{SL}$ and $GDF$ pairs within the expected paleoclimate range that produce an equilibrium ice margin in the Quinault valley at the position of the last glacial maximum terminal

moraine. Over this same set of climate conditions, the Elwha glacier showed extremely variable behaviour. Within the Quinault equilibrium climate range, a 1°C change in temperature corresponded to an expansion of nearly 40 km in the Elwha glacier. Further investigation revealed that this behaviour was related to the valley geometry of the Elwha basin, with the glacier experiencing an ice-elevation feedback as it advanced into the narrowing section of the valley and thickened. This ice-elevation feedback occurs in both the Quinault and Elwha basins, but is more pronounced in cooler and drier climates. In general, our

model was able to reproduce the observed extent of glaciers in warmer climates with higher rates of precipitation. The occurrence of an ice-elevation feedback in both basins, which was linked to the valley geometry, highlights the importance of considering local topography in studies of glacier dynamics.

Our model focused on glacier asymmetry due to the asymmetry in precipitation, and did not consider differences in the efficiency of melting across the Olympic mountains. Our parameter tests suggest that if melt efficiency was even slightly

greater in the Elwha relative to the Quinault, it could have been a significant factor the limiting growth of the Elwha glacier during the last glacial episode. Further investigation with the flowline model could examine the impact of spatial variability in glacier melting.

Overall, we find that spatially variable precipitation influences glacier extent, and that the sensitivity of glaciers to climate varies in space for the Olympic Mountains. This conclusion has implications for other regions with spatial gradients

in precipitation including, the Andes Mountains, the European Alps, the Southern Alps, and Central Asia. Specifically, we suggest that glacier sensitivity to climate likely varies across precipitation gradients forced by topography. Furthermore, the local spatial variability in the relationship between glaciers and climate complicates a prediction of the response of glaciers to ongoing anthropogenic climate change as small changes in climate may have different impacts for different glaciers within the same mountain range. Considering the complex interplay between local topography and climate, it is necessary to take into

account these spatial gradients in future climate projections for accurate predictions of individual glacier behaviour in

mountainous regions.

## Code Availability

A MATLAB version of the glacier flowline model can be accessed through a hydroshare resource:

https://www.hydroshare.org/resource/fe28143081434b0d90f8cffc88e1bfff/ (last access: 10 February 2023).

## Data Availability

Data related to the climate analysis of the Olympic Mountains can be accessed through a hydroshare resource:

https://www.hydroshare.org/resource/0ae232525f984007ba96c1762f21dd3d/ (last access: 10 February 2023).


## Author Contributions

AM and AA wrote the paper. AA initiated the study. RC supplied WRF model output and developed code to analyse

atmospheric weather balloon soundings. GR supplied the flowline model code. AM and AA performed the analysis of the

climate and performed the flowline model simulations. All authors checked and revised the text and the figures of the paper

and contributed to the ideas developed in this study.

## Competing Interests

The authors declare that they have no conflict of interest.

## Acknowledgements

We are grateful to Cristian Proistosescu and [BLANK] for their constructive comments on this manuscript.



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
