# Peer review of "Spatially coherent variability in modern orographic precipitation produces asymmetric paleoglacier extents in flowline models: Olympic Mountains, USA"

_Earth Surface Dynamics, 2023_

## Referee Comment (RC2)

[referee-annotated manuscript omitted]

---

## Author Comment (AC1)

[Figure]

*Figure 1. Panel A: Location of the study area in Washington State, U.S.A. Panel B: Topography of the Olympic Peninsula. Terminal moraines of the west side valleys shown in black (Marshall, 2013; Staley, 2015). Radiosondes are routinely launched from the black star at Quillayute (Rutledge and Ciesielski, 2016). Humptulips (black star) is the location of the paleoclimate record we use. We define the Elwha and Quinault basins as the regions upstream of USGS stream gauges, shown as yellow stars. Panel C: Annual climatological precipitation from the PRISM model (Daly et al., 2008). Panel D: Total modelled precipitation during the OLYMPEX campaign in the winter of 2015-2016 (Skamarock et al., 2008).*

---

## Author Response (AR1)

Dear Reviewer #1,

Thank you for the positive and constructive review. I believe that we edit the paper to address your concerns and make a better product.

Major comments

The abstract is too long, please shorten it. See also l. 19 minor comments below.

I have shortened the abstract significantly so that it is now 363 words long and much more concise. I could not find an abstract length requirement for ESD, but believe that the revised abstract is no longer unusually long or overly wordy.

In Figure 1, please provide an inset with a location in North America. Please make the units consistent (preferable mm per year). Colors in legends do not conform to colors in figures over land, probably owing to the colors being transparent to show the background topography. Please add more labels to the legends. Please add other names used in the text (e.g. Humptulips, Hoh, Queets, Quillayute) and the present extent of glaciers.

I have revised figure one to include a location map showing the study area in North America as a first panel. The units (and colors) are identical for the two precipitation data sets shown in the figure. The legend better reflects the colors with the topographic hillshade over them. I have added modern glaciers and additional names of locations to the revised figure.

l. 46: "Spatial variability in precipitation due to topographic forcing (i.e., a rain shadow) is a well-documented and generally well-understood example of a robust spatial gradient in climate (Roe, 2005)." Under these conditions (windward orographic precipitation) the foehn effect will create a simultaneous temperature gradient, further worsening the conditions for lee-side ice formation. Should this not be part of the research question?

Thank you for mentioning a factor we hadn't considered. The foehn effect is observed in the Cascade mountains to the east of the Olympics – locally called the Chinook wind. However, the impact of the foehn effect for the Olympic Mountains would be quite difficult to assess. One issue is that air flow does not always go over the Olympics; it is also observed to split and be diverted around the range (e.g., Houze Jr, R.A., McMurdie, L.A., Petersen, W.A., Schwaller, M.R., Baccus, W., Lundquist, J.D., Mass, C.F., Nijssen, B., Rutledge, S.A., Hudak, D.R. and Tanelli, S., 2017. The olympic mountains experiment (OLYMPEX). Bulletin of the American Meteorological Society, 98(10), pp.2167-2188.). It would not be reasonable to assume a consistent occurrence or magnitude for this effect. However, the key point in this comment – that there may be differences in the temperature lapse rate on windward and leeward sides of the range – is certainly important and one we did consider. In the discussion section (paragraph beginning about line 440) we mention a number of factors we don't consider and discuss how they might influence our results. I've added a sentence at the end of this paragraph that specifies how the association between humidity and different lapse rates is related to descending air.

By including streamflow gauge observations of basin runoff, snowmelt becomes an important process that could delay runoff and hence distort R_GAUGE. Please comment or briefly discuss in methods section 2.1.3.

We agree and that's why we only calculate R_GAUGE over annual timescales of water years. We added a sentence ~line 200 to make this more explicit.

l. 437: "All precipitation was considered snow." But not at temperatures well above freezing I assume?

We did assume that all precipitation in the model was snow. We argue that this is not unreasonable given the simplicity of the model and the climate of the Olympic Mountains during the last glacial episode. The sea level temperatures in the model are summer temperatures. When we account for a seasonal temperature change (~ 10-15 degrees) it is reasonable to assume that temperatures are below $0^\circ$ Celcius at or near sea level in the winter. This makes it reasonable to assume that cool-season precipitation is snow. In the present climate ~80% of precipitation in the Olympics occurs in the cool half of the year (Nov-March). Thus, it is reasonable to assume that most of the annual precipitation flux is snow – especially for the area above the equilibrium line in the model where there is net gain of mass in the glacier. The model uses an annual mass balance – so we calculate the total accumulation in a year (based on precipitation dominated by snow) and the total ablation (assumed to be controlled by summer temperatures). This simple approach is reasonable given the lack of detailed constraints on past climate and sufficient to explore our research question – i.e. what is the impact of spatial variability in precipitation on past glacier extent.

Minor and textual comments

l. 19: The names 'Elwha' and 'Quinault' need an introduction/explanation. For the uninitiated, these names do not mean much. It makes the presented results in the abstract hard to follow.

Thanks for this comment – we've been thinking about this area for so long that we didn't consider that our audience needs more context! I have tried to revise the abstract to give some more context to the place names while also making the abstract shorter. More place names on Figure 1 may also help in this regard.

l. 78: storm tracks coming from the Pacific Ocean -> extending from the Pacific Ocean (?)

I've replaced that word as suggested.

l. 79: "Winds directed into the topography induce standing waves in the atmosphere that cause lifting upwind of the topography. As air rises, it cools causing condensation of cloud droplets; eventually, their coalescence forms falling hydrometeors." This is unnecessarily long, consider using "When humid air masses are forced over the topography, this results in significant orographic precipitation on the windward side."

I have shortened this description somewhat, but want to retain a little more detail as it is important for considering issues such as a difference in lapse rates on the windward vs. lee.

l. 89: colder -> lower

I've replaced that word as suggested.

l. 90: "Sea surface ... today." Please provide a reference.

I've added a reference as suggested

l. 95: "Angeles Point 7.5-minute Quadrangles" Is something missing here?

This information doesn't need to be in the paper – there are published surficial geologic maps that we cite regarding the extent of past glaciation. This phrase has been removed.

l. 131: " There is strong evidence in the Olympic Mountains for 1) the modern precipitation gradient, and 2) large glaciers that were present on the windward side of the range but not on the lee side during the last glacial episode. " This sentence is unclear. Strong evidence that during the last glaciation, the precipitation gradient was similar to today?

The goal of the sentence was to say that the modern climate gradient is well-established and the past glacier asymmetry is also well established. Our hypothesis is that the modern precipitation gradient also operated in the past and explains the observed past glacier asymmetry. We've reworded this to hopefully be clearer.

l. 168: designed validate -> designed to validate

Thank you, we have fixed the typo.

l. 280: warmer -> higher

I've replaced that word as suggested.

l. 282: slower -> lower

I've replaced that word as suggested.

l. 352: Km -> km

Thank you, we have fixed the typo.

l. 441: Clouds do not condense and evaporate, they form and dissipate...

I've replaced that words as suggested.

Response to Reviewer 2

Thank you for the constructive review.

Major comments:

For me the main limitation of the study is that there is a focus solely on total precipitation, while no separation between snowfall and rainfall is made. This is problematic, because snowfall is the main precipitation component that affects glacier mass balance, whereas mass balance does hardly depend on rainfall (although some of it may refreeze). It would have been more interesting to concentrate on snowfall rather than precipitation gradients between the Quinault and Elwha catchments, especially since it is shown that the precipitation gradient weakens for lower temperatures (Fig. 3a). It is hence likely that the snowfall gradient is weaker than the precipitation gradient, which may also be an explanation for the fact that glaciers currently exist in Elwha but not in Quinault.

We assume that all precipitation in the model was snow. We argue that this is not unreasonable given the simplicity of the model and the climate of the Olympic Mountains during the last glacial episode. The sea level temperatures in the model are summer temperatures. When we account for a seasonal temperature change (~ 10-15 ºC) it is reasonable to assume that temperatures are below 0ºC at or near sea level in the winter. This makes it reasonable to assume that cool-season precipitation is snow. In the present climate ~80% of precipitation in the Olympics occurs in the cool half of the year (Oct-March). Thus, it is reasonable to assume that most of the annual precipitation flux is snow – especially for the area above the equilibrium line in the model where there is net gain of mass in the glacier. The model uses an annual mass balance – so we calculate the total accumulation in a year (based on precipitation dominated by snow) and the total ablation (assumed to be controlled by summer temperatures). This simple approach is reasonable given the lack of detailed constraints on past climate and sufficient to explore our research question – i.e. what is the impact of spatial variability in precipitation on past glacier extent. There does appear to be a tendency for a flatter precipitation gradient in colder events in the modern climate and if that was the case during the cooler conditions when a large glacier grew in the Quinault valley, then it makes the small Elwha glacier less easy to explain by the precipitation gradient alone – suggesting a more important role for differences in the ablation side of the mass balance between the two basins. Our work documenting the modern climate gradients suggests that the gradient is unlikely to have been steeper during the last glacial maximum. Our models suggest that if the paleoclimate was relatively warm and maintained these gradients, they would be sufficient to explain the observed glacier asymmetry. If the precipitation gradient were steeper (or the paleoclimate was on the cooler end of allowed) then the asymmetry probably requires some differences in mass loss between the two basins in addition to the mass gain we focused on. I propose to add a sentence to the paragraph beginning about line 440 that acknowledges the effect of weaker past precipitation gradients.

.

It is quite interesting that there currently are no glaciers in the Quinault valley, while they do exist in the drier and warmer Elwha catchment. Furthermore, the Quinault catchment has elevations that could be high enough for glaciers to form (the highest elevations are not much lower than the Elwha catchment if I look at Fig 1a). With temperature and precipitation being the main drivers of glacier mass balance, it seems very unlikely that a positive mass balance that is needed for a glacier to form would apply in Elwha basin and not at some high elevation site in Quinault basin. It would be really helpful to include a map of present glacier extent in Fig. 1. What other factors could affect the glacier distribution? Could wind driven snow redistribution be a significant factor, i.e. leading to net erosion of snow in Quinault and net deposition in Elwha? Does the orientation of the grid matter with more north-facing slopes in Elwha than Quinault, affecting solar radiation absorption? Or maybe the spatial precipitation is simply weaker than expected?

Thank you very much for this excellent question! It led us to realize that we made a mistake in stating there are no glaciers in the Quinault valley at present. We've been so focused on the past we neglected to look at the modern glaciers in these valleys. To correct this oversight, we downloaded the GLIMs glacier database (GLIMS Consortium, 2005. GLIMS Glacier Database, Version 1. Boulder Colorado, USA. NASA National Snow and Ice Data Center Distribute Active Archive Center. DOI: https://doi.org/10.7265/N5V98602 [4/15/23] and overlaid it on our study area. We include these outlines of modern glaciers in revised versions of Figure 1a and Figure 5. That said, they're quite small. In the Quinault basin, the entire glacier area is less than 1 $km^2$ as of 2017. The largest single glacier in the Quinault basin was ~0.4 $km^2$ in area as of 2017. In the Elwha valley, the total glacier area is about 2 $km^2$ as of 2017 and the largest glaciers (the Carrie Glacier ~0.4 $km^2$ as of 2017 and the Fairchild Glacier ~0.25 $km^2$ as of 2017) are hosted on N/ENE sides of Mt. Carrie in the Bailey Range which is about 11km north of the highest peak in the Olympics and along the western edge of the Elwha basin about 30km north of the headwaters. We have updated the manuscript to correct this mistake. We also cite a paper by Fountain et al. that was published after Margason had completed his MS project that became this manuscript.

To address the larger questions you pose, it is likely that wind blown snow contributes to the larger Elwha valley glaciers as well as shading in the north-facing cirques. Measurement of snow is a challenge – especially in these remote alpine areas. The differences in river discharge across the range are similar to precipitation gradients, which suggests that the gradient can not be changed enormously by blowing snow in the present. Past snow redistribution may have been more effective, in which case the glacier asymmetry might require differences in melt efficiency as well as the precipitation gradient.

A final major comment is on the ice flow modelling, which only considers one flowline, whereas based on the map in Figure 5 it seems that several glacier flowlines may have existed and merged into one glacier tongue in both valleys. Ignoring these potential

tributary glaciers may lead to a severe underestimation of glacier length in the simulations. The impact of this omission should be discussed and it should be mentioned that this adds to considerable uncertainty in the results of the climate sensitivity analysis.

We do only consider a single flowline and include the contributions from tributary valleys by changing the basin width along the flowline. Thus, the integrated area above each point on the flowline is similar to that in the real landscape, and, we argue this is giving us the first-order impact of the presence of tributary valleys/multiple flowlines.  This simplification does not allow us to consider the case of a single subbasin creating an independent ice lobe and we assume climate only varies along the length of the flowline – i.e. we don't include spatial differences in precipitation perpendicular to the flowline from different subbasins. We have also used a 2-d model, PISM, on the Olympic peninsula and our initial results there – a tendency for a large Elwha glacier to form – led us to step back from the added complexity to explore the impact of a simplified precipitation gradient on flowlines – presented here – as a way to understand the broader tendencies of the system.  I have added a couple sentences to the end of the paragraph starting at about line 470 to describe the benefits of a two-dimensional model.

Minor & textual comments:

See annotated pdf.

Line 16 – replaced meters with meters water equivalent

Line 23-24 was cut to shorten the abstract

Line 26 phrase was cut to shorten the abstract

Line 90 – this is precipitation near the paleoclimate study site – clarified in revision

Line 93 – the glaciers likely advanced and retreated multiple times during the Wisconsin episode – leaving behind recessional moraines.  We've clarified the language.

Paragraph centered on line 105 – This is the discussion of past glaciers – and we've corrected our mistake about the modern distribution of ice.  The potential importance of windblown snow is also mentioned here and we cite recent work on glacier extent and loss in the Olympics.

Line 163-4 We reworded it this way to clarify:

We average the PRISM estimate of precipitation across each basin to calculate $P_e$ and $P_q$ and define a rain-shadow index, $R_{PRISM}$, as the ratio of the $P_e$ and $P_q$

Line 168 – corrected typo

Line 208 – clarified to indicate that the precipitation is averaged in space across each basin before calculating the ratio

Table 1 – We agree – precipitation is very similar in the two basins right at the drainage divide.  R is calculated as the spatial average across the basins – so the large value of RWRF reflects the pronounced drying in the eastern portions of the Elwha basin.

Line 245 – Measurement of precipitation in mountains is difficult, in general, and measurement of snow is more difficult than measurement of rain. The PRISM model does not distinguish between rain and snow. This makes constraining a difference in modern precipitation for rain vs. snow challenging.

During the cooler conditions of the last glacial maximum, we argue that the majority of precipitation was snow.  In the modern climate, an increasing fraction of the precipitation is rain at the surface. That said, the rain falling in the Olympics, especially in the winter season, likely began as ice in the atmosphere and had a fall speed (and downwind advection) identical to precipitation that falls as snow during descent. Therefore, we expect that the gradient in precipitation reaching the surface is not greatly changed by a phase change near the surface. Later redistribution by wind is possibly for snow, but not for rain and this factor could be important in advecting moisture short distances over the drainage divide. Some weather forecast model results suggest that precipitation gradient in the modern may be flatter for snow-dominated events. Both snow advection and flatter precipitation gradients would make it less likely that precipitation patterns alone can explain the difference in LGM glacier extent. We've edited the discussion section to explicitly say this (line ~469).  It fits with the overall finding of the paper – that precipitation gradients may be sufficient (if we're at the warm end of the paleoclimate limits) but that other factors (esp. related to mass loss) may be necessary.  One key finding is that it is unlikely that there were stronger precipitation gradients in the past – so we are able to place a constraint there.

Line 280 – There is definitely still a rain-shadow in the winter. The PRISM model is based on existing gauge data and shows a clear difference from W to E. Precipitation is dominated by winter events (roughly 80% of the precipitation is in the cool season) – so the gradient in PRISM reflects the occurrence of a rain shadow during winter.

Lines 291-296 – Thanks for this comment!  This analysis was motivated by the finding of Siler et al. of correlations between rain shadow strength and climate indices in the Cascade Mountains to the east of the Olympics – so we expected we might find these relationships.  We neglected to explain that context to the reader and propose to add that in the paragraph starting around line 234.

Line 313 – Based on this comment and the previous one regarding different precipitation gradients for snow vs. rain, I have added a couple sentences in the introduction that emphasize the contribution from snowfall in the modern climate and the dominance of stable conditions and orographic enhancement of stratiform precipitation – all of which indicates that snowfall gradients are likely very similar to rainfall gradients.

Line 338 – We change the basin width to account for area above each point on the flowline – so this incorporates the influence of the accumulation area added by tributary glaciers.

Line 347 – This should be summer temperature – we've clarified in the revision.

Line 363 – We argue that at LGM conditions the majority of the precipitation is likely to have been snow.

Line 369 – The melt factor chosen comes from previous work on Mount Baker – which is in the Cascade Mountains just east of our study area.

Line 377 – Ice is required to start because of how mass flux at the glacier terminus is handled.

Line 380 – I'm happy to move this to a data availability section if that's more appropriate.

Line 399 – I'm not sure I understand the suggestion here.  We didn't study any cases on flat topography – is the idea simply that the ice-elevation feedback is more significant in those cases?

Line 417 – Yes – this is also a method – yet the context for why we did it really requires consideration of the main results.  So we made a choice to explain these limited other models here rather than presenting the first part of this paragraph (which is really a discussion of results) in the previous section. I can move the entire paragraph to the previous section to make the compromise in a different way.

Line 420 & 423:  The idealized geometry we use for the Quinault neglects the highest part of the basin – this is a narrow valley about 6.5km long and about 15 km$^2$ in area. The entire basin along the flowline is about 425 km$^2$, so we're neglecting 3% of the area by making this simplification instead of including an additional valley segment above the broad headwaters of the Quinault. This simplification is reasonable when considering the equilibrium for the large LGM glacier here – which is the main focus of our work. However, I think it probably is the dominant factor controlling why we don't

produce an equilibrium small glacier in this valley – that small glacier would be within the omitted part of the basin. We could certainly add this to the Quinault geometry and run additional models, but we're really making a quite minor point here and I think it's probably better to just leave it out entirely. The bigger idea of this paragraph is simply that the geometry of the Quinault is broadly similar to that of the Elwha and an ice-elevation feedback as the glacier advances through the narrow section of the valley is also observed here.  As for the impact of rain vs. snow – our failure to find an equilibrium small glacier when we assume all precipitation is snow doesn't seem to suggest that if we accounted for some precipitation being rain we might find such an equilibrium.

Line 438 – Yes – this sentence is misleading and will be corrected to reflect the lack of contribution of rain to mass gain.

Line 449- We now describe where the melt factor comes from – a neighboring region.

Line 451 – You're right we were comparing with degree day factors. I propose to remove this sentence from a revised manuscript

Line 484 – The lower gradient parts of the record are typically not entire storm events – they are lighter precipitation that occurs only during a portion of the event and, therefore, we don't conclude that we could have an entire climatology driven by these pieces. We can't have post-frontal precipitation without the frontal precipitation. We do try to convey the uncertainty – which is not insignificant – but we were really looking for evidence of different types of behavior that might have been dominant in the past.  We don't see a lot of variability in the spatial pattern of precipitation from event to event for relatively large events ( see Anders, Alison M., Roe, Gerard H., Durran, Dale R., and Minder, Justin R., 2007, Small-scale spatial gradients in climatalogical precipitation on the Olympic Peninsula, Journal of Hydrometeorology, v.8, 1068-1081. ).  It's possible that events typical of the last glacial maximum simply don't occur in the modern climate, but given the location (downwind of the Pacific Ocean) and typical stable lifting (relatively little convection), we argue this is not likely.

Line 495 – Yes – this is the direction it would go.  However, we argue that the majority of precipitation at LGM was likely snow – and a cooler climate would push toward even greater dominance of snow.

---

## Author Response (AR2)

Thank you for the comments. I have made changes for each comment, as described in detail below, that, I believe, improve the manuscript and address your concerns. Additionally, I have edited the manuscript throughout for units, spacing of units, use of "valley", "basin", and "glacier" with correct capitalization. I have checked for completeness and correctness of citations in text and in the reference list.

Line 20 – I have reworded as suggested. I have revised the sentence starting "In the dry lee," following the suggestion.

Line 27 – Here and elsewhere I have described our model as relating to the glaciation of the valleys rather than the basins as suggested.

Line 29 and 31 – I have specified that these are July average sea-level temperatures.

Line 32 – Yes – this is what I intended – I have revised the sentence to make clear what we intend to say.

Figure 1 – I have added names of countries to panel 1, changed the label in panel B and added a reference to panel d in the text. The references to panels in this figure have been corrected throughout the text.

Line 59 – changed meters to m w. e.

Line 63 – reworded as suggested

Line 73 – changed to 35 ka to follow journal style guide.

Line 75 – units separated from number and millimeters per year abbreviated as suggested.

Line 89 - space added

Line 91 – citation corrected

Line 121 – Episode removed

Line 170 – abbreviation removed

Line 180 reworded as suggested

Line 184 – units abbreviated as suggested

Line 189 – citation corrected

Line 208 – 210 – abbreviations replaced with citation date

Line 231 & 268 – table header moved above table

Line 283-284 – units abbreviated

Figure 3: Revised to add panel letters and caption revised.

Figure 4: Caption revised

Line 328: basins changed to Valleys

Table 3: Header moved and headings edited as suggested

Figure 5 – legend edited as suggested.

Line 371: degree symbol added.

Line 379: The units are correct as the sliding parameter is multiplied by the driving stress cubed over the ice thickness to get a velocity. I have added the reference that Oerlemans refers to as well.

Line 388: Changed as suggested – we meant local maximum extent but the suggested wording is more clear.

Figure 6 – formatting of units corrected.

Line 443 – changed wording to "during the early Wisconsin glacial advance"

Line 456-7 : units corrected

Line 500 – units spaced correctly

Line 605 – this is a typo – it's a partial repeat of the following reference to Hobbs, 1978

Line 623 – corrected typos

Line 679 – corrected

Line 704 – doi added

Line 706-7 words in title capitalized

Line 720 and on – corrected here and in body

Line 734 - corrected

---

## Author Response (AR3)

Dear ESPL,

The only change I have made to the accepted manuscript is to include the DOI numbers for the model code and datasets which are now published with fixed DOIs.

Sincerely,

Alison Anders